# Short-term effects of GPS collars on the activity, behavior, and adrenal response of scimitar-horned oryx (*Oryx dammah*)

Jared A. Stabach[1]*, Stephanie A. Cunningham[1¤a], Grant Connette[1], Joel L. Mota[1¤b], Dolores Reed[1], Michael Byron[1¤c], Melissa Songer[1], Tim Wacher[2], Katherine Mertes[1], Janine L. Brown[1], Pierre Comizzoli[1], John Newby[3], Steven Monfort[1], Peter Leimgruber[1]

1 Smithsonian National Zoo and Conservation Biology Institute, Front Royal, VA, United States of America, 2 Zoological Society of London, London, England, United Kingdom, 3 Sahara Conservation Fund, Bussy Saint Georges, France

¤a Current address: Department of Environmental and Forest Biology, State University of New York College of Environmental Science and Forestry, Syracuse, NY, United States of America
¤b Current address: Davis College of Agriculture, Natural Resources & Design, University of West Virginia, Morgantown, WV, United States of America
¤c Current address: North Carolina State University, Raleigh, NC, United States of America
* stabachj@si.edu

**Data Availability Statement:** All relevant data are within the manuscript and its Supporting Information files.

## Abstract

GPS collars have revolutionized the field of animal ecology, providing detailed information on animal movement and the habitats necessary for species survival. GPS collars also have the potential to cause adverse effects ranging from mild irritation to severe tissue damage, reduced fitness, and death. The impact of GPS collars on the behavior, stress, or activity, however, have rarely been tested on study species prior to release. The objective of our study was to provide a comprehensive assessment of the short-term effects of GPS collars fitted on scimitar-horned oryx (*Oryx dammah*), an extinct-in-the-wild antelope once widely distributed across Sahelian grasslands in North Africa. We conducted behavioral observations, assessed fecal glucocorticoid metabolites (FGM), and evaluated high-resolution data from tri-axial accelerometers. Using a series of datasets and methodologies, we illustrate clear but short-term effects to animals fitted with GPS collars from two separate manufacturers (Advanced Telemetry Systems—G2110E; Vectronic Aerospace—Vertex Plus). Behavioral observations highlighted a significant increase in the amount of headshaking from pre-treatment levels, returning below baseline levels during the post-treatment period (>3 days post-collaring). Similarly, FGM concentrations increased after GPS collars were fitted on animals but returned to pre-collaring levels within 5 days of collaring. Lastly, tri-axial accelerometers, collecting data at eight positions per second, indicated a > 480 percent increase in the amount of hourly headshaking immediately after collaring. This post-collaring increase in headshaking was estimated to decline in magnitude within 4 hours after GPS collar fitting. These effects constitute a handling and/or habituation response (model dependent), with animals showing short-term responses in activity, behavior, and stress that dissipated within several hours to several days of being fitted with GPS collars. Importantly, none of our analyses indicated any long-term effects that would have more pressing animal welfare concerns.

**Funding:** Funding for this project was provided by the Environment Agency - Abu Dhabi, the Sahara Conservation Fund, and the Smithsonian Conservation Biology Institute.

**Competing interests:** The authors have declared that no competing interests exist.

## Introduction

Global Positioning System (GPS) devices have revolutionized the field of animal ecology [1–3], providing detailed information about how animals move and utilize space across diverse and often rapidly changing landscapes (e.g., [4–7]). A variety of taxa can now be monitored, ranging in size from pint-sized ovenbirds (*Seiurus aurocapilla*; [8]) to multi-ton elephants (*Loxodonta africana; e.g.*, [9]). In some instances, animals are now being monitored over their entire lifespans [10], a result of technological innovations (e.g., solar rechargeable batteries) or re-tagging efforts, and at temporal resolutions (minutes to seconds) that would have been unimaginable just a few decades ago.

Since the inception of animal tracking, scientists have recognized the ethical concerns of fitting animals with tracking devices, providing general guidelines (e.g., devices should weigh <5% of total animal body weight) and in most cases, requiring tracking studies to undergo a thorough review from animal care and use committees before being initiated [11]. While various studies investigating potential adverse effects resulting from animals wearing telemetry devices exist (for a review, see [12]), most mammalian studies (~85% reported in [12]) have focused on small- to medium-sized animals (< 15 kg), which are easier (and less expensive) to monitor and manipulate in laboratory settings than studies on large animals (although see [13–17]). Prevailing results of these studies illustrate that the effects of fitting tracking devices on animals are generally minimal, although severe and study-specific impacts have been reported [12,18]. And while research into the effects on large mammals do exist (see [13,15–17,19]), these studies are mostly focused on the effects that result from capture and/or chemical immobilization, rather than the impact of collars themselves.

The most common telemetry device for monitoring large mammals over extended time periods is the GPS collar. Designed to fit around an animal's neck, devices are comprised of a satellite receiver that points directly upward to communicate with the Navstar satellite constellation and a battery pack located underneath the animal's neck which also serves as a counterbalance to keep the device in place. Fitting a tracking device around an animal's neck, however, has inherent risks ranging from mild irritation [20] to severe tissue damage [21,22], reduced fitness, and death (either by increased predation pressure or as a direct result of the device). Behavioral and habitat use changes can also occur [23], biasing study results and leading to losses in data integrity [3,24].

Zoos and captive breeding facilities offer an opportunity to monitor large animals in controlled settings, providing valuable information that may benefit wild populations. The purpose of our study was to evaluate the impact of GPS collars fit on captive-bred scimitar-horned oryx (*Oryx dammah*, hereafter oryx). Once widespread across Sahelian grasslands stretching from Senegal to Sudan, the species is now extinct in the wild due to overhunting and habitat loss [25]. Some 5,000–10,000 captive-bred oryx exist in institutions globally, the descendants of oryx taken from their native habitat in the 1960s. A consortium of institutions led by the Environment Agency–Abu Dhabi, the Government of Chad, and the Sahara Conservation Fund have embarked on an ambitious initiative to reintroduce oryx to a portion of the animals' former range in Chad. Since August 2016, 194 oryx have been reintroduced to the Ouadi Rimé-Ouadi Achim Wildlife Reserve, the second largest terrestrial protected area in Africa (77,360 km$^2$) and the heart of the species' former range. Nearly every individual reintroduced to the reserve has been or will be fit with a GPS collar, the only economically viable option to monitor movement, survival, and provide a means for assessing reintroduction success across this large, inaccessible region. Given the species' conservation status [25], evaluating potential adverse effects resulting from GPS collars was a research priority.

With the advent of advanced laboratory techniques to assess subtle changes in adrenal activity and bio-logging devices to measure activity in micro-second intervals, we now have the ability to more thoroughly evaluate the response of animals to GPS collars. We used three methods, including traditional behavioral scans, assessments of fecal glucocorticoid metabolite (FGM) concentrations, and analyses of high-resolution data from tri-axial accelerometers, to assess potential adverse effects of GPS collars fitted on oryx. We predicted that animals would acclimate to GPS collars within a few days of being handled/fit, resulting in a short period in which irritation behaviors (i.e., head shaking) and hormonal stress levels (i.e., FGM) increased before returning to pre-collaring levels. By incorporating multiple analysis methods, we provide detailed information on the short-term effects of devices that have proven to be essential for reintroduction/relocation efforts, and which should provide beneficial in discussions with the general public on the use of these devices in wildlife studies.

## Materials and methods

### Study area

Research was conducted at the Smithsonian Conservation Biology Institute in Front Royal, Virginia, USA (38˚53' N, 78.9'W). Habitat across the 1,200 ha facility is predominantly Appalachian oak forest [26], consisting of managed pasture for research and husbandry purposes. Oryx were separated into multiple enclosures to control reproduction and reduce competition-related injuries. Female oryx (11 adult, 1 juvenile) were housed together as one group in a ~1.5 ha enclosure, inclusive of fenced pasture and a barn facility where veterinary procedures were conducted. Intact (i.e., non-vasectomized) males (3 adults, 0 juvenile) were located in separate, unconnected enclosures > 1 km from females.

### Study design and data collection

Thirteen (13) oryx were selected for inclusion in this study. Animals fitted with GPS collars ($n$ = 10) were between 1 and 16 years of age. No animal had ever before been fitted with a GPS collar. Control animals ($n$ = 3), used as a comparison dataset for analyses of FGMs, were 7 to 18 years old (Table 1). All aspects of animal handling were administered by Smithsonian staff veterinarians and approved by the Smithsonian National Zoo and Conservation Biology Institute's Institutional Animal Care and Use Committee (Proposal #15–32).

Female oryx were physically restrained to fit GPS collars using a hydraulic restraint device (TAMER®, Fauna Research Incorporated, Red Hook, NY). No chemical immobilizations were used. Handling activities occurred at the same time animals were being prepared for artificial inseminations. Each female received two injections of cloprostenol sodium 11 days apart (500 µg, IM; Estrumate, Intervet Incorporated, Summit, NJ) to synchronize estrus [27]. These procedures provided our team with two opportunities to handle each animal and evaluate collar fit. A third opportunity to evaluate collar fit was presented at the time of artificial insemination (56 h after the second prostaglandin injection). All female oryx were weighed throughout the study period (Table 1). GPS collars were fitted on male oryx at the same time semen was collected. Males were immobilized with a mixture of etorphine, medetomidine, and ketamine. No injuries occurred during handling and/or collaring.

Global Positioning System (GPS) collars from two manufacturers (Advanced Telemetry Systems [ATS Incorporated, Isanti, MN] and Vectronic [Vectronic Aerospace GmbH, Berlin, Germany]), were tested on oryx: five ATS Iridium/GPS collars (model G2110E) and five Vectronic Iridium/GPS collars (model Vertex Plus). Collars weighed 1.1 kg and 0.6 kg, respectively, ≤ 1% of oryx body weight (Table 1). Collars were fit on 22 October (6 females), 02 November (2 females), and 03 November 2015 (2 males). Collars were programmed to

**Table 1. Summary of scimitar-horned oryx (*oryx dammah*) included to evaluate short-term effects of GPS collars.** Animals (ID) were fit with Advanced Telemetry System (ATS; model G2110E) and Vectronic Aerospace (Vectronic; model Vertex Plus) GPS collars. Control (non-collared) animals also listed (Collar Type: None). Change (Δ) in body weight compares animal weights on 03-Nov-2015 (Weight) and 15-Dec-2015. Percent (%) body weight of GPS collars based on collar weights of 1.1 Kg (ATS) and 0.6 Kg (Vectronic). Checkmarks (✔) indicate animal inclusion in study components. (-) Indicates data not collected.

| ID | Sex | Age (years) | Collar Type | Weight (Kg) | % Body Weight[a] | Δ in Body Wgt (Kg) | GPS Monitoring Period (Days) | Behavior Observations | FGM[c] | Accelerometer Data |
|---|---|---|---|---|---|---|---|---|---|---|
| 114531 | F | 5 | ATS | 154.2 | 0.73 | -5.4 | 40 | ✔ | ✔ | |
| 114542 | F | 5 | ATS | 124.3 | 0.89 | 0 | 39 | ✔ | ✔ | |
| 114842 | F | 4 | ATS | 108.9 | 1.03 | -4.6 | 40 | ✔ | | |
| 114843 | F | 4 | ATS | 115.2 | 0.96 | -1.8 | 29 | ✔ | ✔ | |
| 115093 | F | 1 | ATS | 115.2 | 0.95 | 0 | 46 | ✔ | ✔ | |
| 113469 | M | 16 | Vectronic[b] | - | - | - | 27 | ✔ | ✔ | |
| 114426 | F | 8 | Vectronic | 130.6 | 0.46 | -1.8 | 40 | ✔ | ✔ | ✔ |
| 114839 | F | 3 | Vectronic | 133.4 | 0.45 | 1.8 | 29 | ✔ | ✔ | ✔ |
| 114915 | M | 8 | Vectronic | - | - | - | 28 | ✔ | ✔ | ✔ |
| 114969 | F | 10 | Vectronic | 136.1 | 0.46 | -11.8 | 40 | ✔ | ✔ | ✔ |
| 113204 | M | 18 | None | - | - | - | - | | ✔ | |
| 114427 | F | 7 | None | - | - | - | - | | ✔ | |
| 114968 | F | 10 | None | - | - | - | - | | ✔ | |

[a]Percent body weight of GPS collar

[b]Accelerometer data corrupt/excluded from analysis

[c]Fecal Glucocorticoid Metabolites

automatically drop-off animals or were manually removed for further behavioral investigations (Cunningham et al. In review). Total time oryx were collared ranged from 27 to 46 days (Table 1).

We split the study into three periods (pre-treatment, treatment, and post-treatment) to evaluate the length of time that short-term behavioral and adrenal hormone effects associated with fitting animals with GPS collars were present. The pre-treatment period represented the period prior to animals being fit with collars (< 0 days of the collar fitting date). The treatment period represented the period in which oryx were fitted with GPS collars and the time period in which we expected to observe short-term adverse effects (0–3 days after collar fitting). The post-treatment period was the time period in which we expected short-term effects to have subsided (> 3 days of the collar fitting date). GPS collars remained on animals throughout the post-treatment period. Treatment periods, however, represent relative dates since collar fitting occurred on different dates. We shifted the treatment periods +1 day (as described below) when analyzing fecal glucocorticoids due to the recognized lag in detecting circulating hormones in fecal material [28–30].

All statistical analyses for this study were formulated in a Bayesian framework. Models were fit using Markov chain Monte Carlo (MCMC) simulation in JAGS [31], executed via the jagsUI package [32] in program R [33]. Code to execute all analyses described below are provided in the Supplementary Materials (S1 Code: Behavioral Observation Analyses; S2 Code: Fecal Glucocorticoid Metabolite Analyses; S3 Code: Accelerometer Data Analyses). Each File includes code, data, details of the priors, and specifics of the simulations.

## Behavioral observations

**Data collection and processing.** Behavioral data were collected by one observer (JM) between 08 October 2015 and 17 November 2015. Each animal (*n* = 10) was observed during

each of three treatment periods (Pre-treatment, Treatment, and Post-treatment). Animals were acclimatized to the presence of the observer for two weeks prior to the data collection period, 20–30 minutes each day. Observations were collected in the morning (female oryx) and afternoon (male oryx). The observer stood outside enclosures, ~20 meters from surveyed individuals. Female oryx were observed simultaneously in groups ranging from 1 to 5 individuals. Male oryx were observed individually.

Behavioral sampling for each individual was conducted across repeated 10-minute observation periods, following details provided by [34]. During each observation period, a behavioral tally was recorded every 15 seconds. Behaviors, derived from [35] and consistent with studies investigating collaring effects [36,37], were divided into six categories (Table 2). These behaviors represented normal oryx activity (e.g., feeding, walking) and those that might indicate physical irritation caused by the collars (e.g., scratching, head-shaking). Behavioral data were collected for a total of 198 observation periods (Pre-treatment (17 days) = 55 periods [5.5 ± 3.1 per individual]; Treatment (4 days) = 30 periods [3.0 ± 0.8 per individual]; Post-treatment (22 days) = 113 periods [11.3 ± 4.7 per individual]). In total, we observed 2215 behaviors during the Pre-treatment period (221.5 ± 124.4 per individual), 1211 behaviors during the Treatment period (121.1 ± 31.6 per individual), and 4544 behaviors during the Post-treatment period (454.4 ± 188.1 per individual).

We predicted that irritation behaviors (e.g., headshaking, scratching) would increase after animals were fitted with collars and subside by the post-treatment period (Pre-treatment < Treatment > Post-treatment). This would indicate short-term effects related to animals being fitted with collars, possibly reflective of a perceived fear of strangulation—similar to a predator attack [36,38,39]. Continued irritation (Pre-treatment < Treatment = Post-treatment) would indicate more severe/longer term animal welfare concerns.

**Statistical analyses.** To assess differences in the frequencies of observed response behaviors between treatment periods (Pre-treatment, Treatment, Post-treatment), we fit a multinomial logistic regression model to the data. The behavioral observations for each 10-minute observation period were summarized as counts indicating the number of times each of the six response behaviors was observed (Table 2). Each vector of behavioral counts was then treated as a multinomial response, where the probability of each response behavior was estimated for each treatment period. We also incorporated a random effect of individual to account for repeated measurements of the same individuals and uneven numbers of observations across individuals in each of the three treatment periods. Statistical significance was assessed by determining whether the 95% credible intervals for the estimated frequency of irritation behaviors from the Treatment and Post-treatment periods (respectively) overlapped the posterior distribution median of the Pre-Treatment period (i.e., control). Further details, including model specifications, can be found in S1 Code.

**Table 2. Response descriptions, derived from Packard et al. (2014), to evaluate behavioral changes related to GPS collar fitting.** Behavior tallies collected every 15 seconds during 10-minute observation windows.

| Behavior | Description | Abbreviation |
|---|---|---|
| Standing (Head Up) | Animal is stationary and/or feeding on food items placed in a raised feeder. Head is higher than the shoulder. | HU |
| Standing (Head Down) | Animal is stationary and/or feeding on food items on the ground. Head is lower than the shoulder. | HD |
| Laying | Animal is stationary with legs folded and body in contact with the ground. | LAY |
| Headshaking | Animal is quickly rotating its head, left to right or forward to backward. | HDSK |
| Scratching | Animal is applying pressure with muzzle, teeth, horns or hooves while moving rapidly over a small area of the body. | SCRATCH |
| Locomotion | Animal is engaged in locomotion, moving from one point to another. | LOCO |

## Fecal glucocorticoid metabolites

**Data collection and processing.** Fecal samples (~20 g samples) were collected between 15 October 2015 and 18 December 2015 (64 days) to evaluate changes in FGM concentrations. Unique sample origin was difficult to determine for female oryx housed together as a herd, resulting in lower sample sizes than males. We collected fecal samples ($n = 76$) from nine animals fit with GPS collars (treatment; $n_t = 55$) and three animals that were not (control; $n_c = 21$), to compare hormonal changes related to handling related activities.

Only moist fecal samples, with no visible signs of urine contamination, were collected. Samples were homogenized during collection by hand to more evenly distribute hormones and decrease sample variability [40] and assayed within 1-month of the last date of sample collection. Samples were placed immediately in Nasco Whirl-Pak™ storage bags and transported in a cooler to a -20°C freezer for further analysis. Frozen fecal samples were lyophilized for 4 days; dried samples were crushed into a fine powder. Approximately 0.1 g of each sample was transferred to a test tube for analysis. FGMs were extracted from each sample using methods developed by [41] and analyzed using a corticosterone-I$^{125}$ radioimmunoassay (RIA) kit (MP Biomedicals, LLC; Santa Ana, CA). The sensitivity of the assay is 1 ng/mL. We validated the RIA by demonstrating parallelism between serially diluted fecal extracts and the standard curve, and significant ($> 90\%$) recovery of exogenous corticosterone standard added to fecal samples before extraction. All samples were analyzed in duplicate with acceptable coefficient of variation values of less than 10%. Intra- and inter-assay coefficients of variation of control samples included in the kit were $< 10\%$ and $< 15\%$, respectively.

Because animals were collared on different dates, we matched the samples to a relative collaring date for each individual ('Day 0'). Data were further subset to the 8 days prior to and 10 days following collaring dates to remove subsequent periods when animals were handled and which may have led to multiple observed adrenal responses. Data were combined across individuals with different age and sex classes due to low sample sizes in order to evaluate overall support for each of the following hypotheses (Fig 1):

1. No Response—Animals exhibit no adrenal response to the collar, resulting in a consistent linear trend in FGM with no change over time;

2. Stress Response—Animals exhibit an adrenal response to the collar, resulting in an increase in FGM after collar fitting that does not subside;

3. Habituation Response—Animals exhibit a habituated response to the collar, resulting in an increase in FGM after collar fitting that declines slowly over time; and

4. Handling Response—Animals exhibit a handling response to the collar, resulting in an increase in FGM after collar fitting that declines quickly (within a few days).

**Statistical analyses.** We used a piecewise regression approach (e.g., [42]) to develop statistical models representing each of these four hypothesized responses of FGMs to GPS collaring (Fig 1). For all models representing an adrenal response (hypotheses 2–4 above), we expected a lag of one day in peak FGM levels after collaring, due to the time it takes hormones to enter the circulatory system and for metabolites to be detected (i.e., gut passage time; [28–30]).

The 'No Response' hypothesis was represented by an intercept-only model where the amount of FGMs ($\beta_0$) remained constant over time. We represented the 'Stress Response' hypothesis by a piecewise regression model with a breakpoint, $k$, signifying the time lag (fixed at 1 day) at which we would expect to observe a change in FGMs after GPS collars were fitted. This model included an initial pre-collaring intercept ($\beta_0$) and an additive change in the

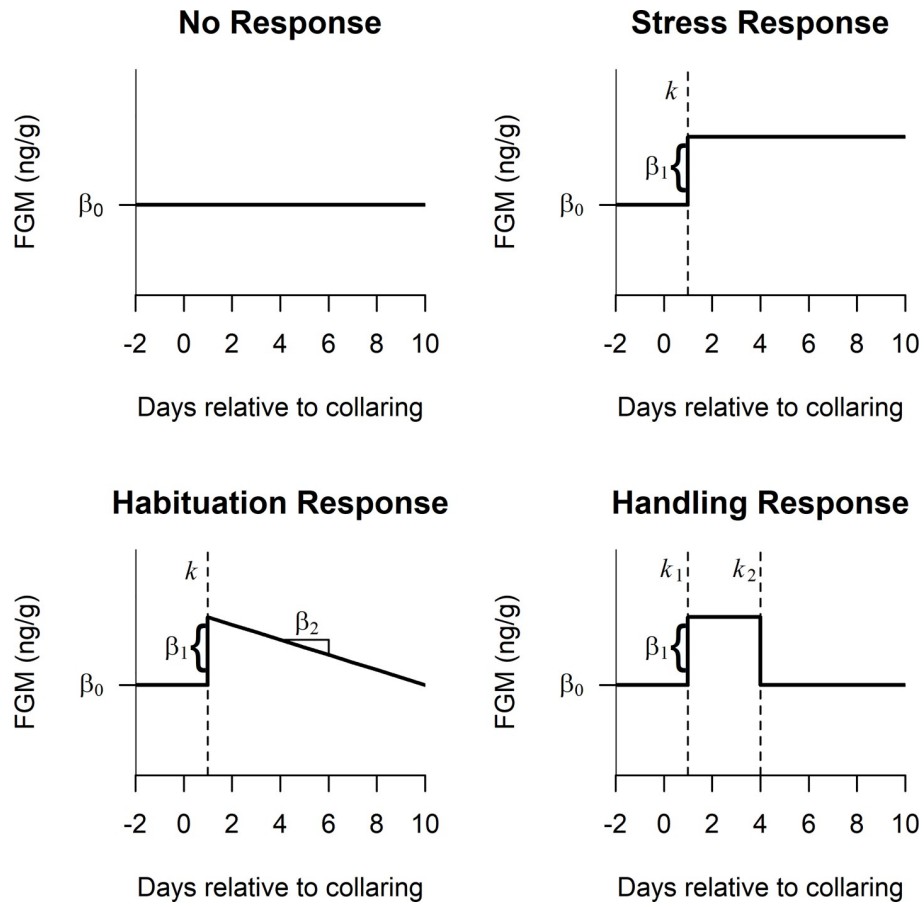

**Fig 1. Hypothesized fecal glucocorticoid metabolite (FGM) (ng/g) adrenal responses to GPS collaring fitting [Day 0 = collar fitting date].** No Response—consistent linear trend with no slope; "Stress" Response—increase in FGM that does not subside; Habituation Response—increase in FGM that declines slowly over time; Handling Response—increase in FGM that declines rapidly. FGM response is time lagged [Day 1, dashed vertical line (k or k1, model dependent)] due to the time in which hormones enter the circulatory system and are metabolized (i.e., the gut passage time; [28–30]).

intercept ($\beta_1$) after the breakpoint. Similarly, the 'Habituation Response' hypothesis was represented by a piecewise regression model with an initial mean FGM level ($\beta_0$), a breakpoint on the first day post-collaring ($k = 1$), and a change in FGMs ($\beta_1$) after the breakpoint. However, this model included a slope parameter ($\beta_2$), reflecting our assumption that FGM levels should decline after the breakpoint and gradually regress to pre-treatment levels. Lastly, the 'Handling Response' hypothesis was represented by a piecewise regression model with two separate breakpoints. The first breakpoint, $k_1$, was fixed to the first day post-collaring ($k_1 = 1$) and the second breakpoint, $k_2$, indicated the end of the handling response and was estimated from the data. Here, $\beta_0$ represents the mean initial FGM level and $\beta_1$ represents the change in hormone levels between breakpoints $k_1$ and $k_2$. Detailed model descriptions are provided in S2 Code.

We fit separate models for control and treatment animals, providing a comparison with animals that were not fitted with a GPS collar but may be affected by disturbances caused during collaring related activities. Best fitting models were identified by performing leave-one-out cross validation, selecting the model that minimized the sum of squared errors.

## Accelerometer data

**Data collection and processing.**  Vectronic GPS collars were equipped with internal tri-axial accelerometers, collecting raw data (8 Hz) of the change of direction in the x-, y-, and z-axes (surge, sway, and heave, respectively). Results were recorded for the duration of the study period. Data from one collar was corrupt and excluded from analyses (Table 1). Advanced Telemetry Systems collars were also excluded from analyses since accelerometers in these collars provided data pre-processed and summarized across each axis in 15-minute intervals, precluding identification of specific behaviors over time.

**Random forests classification.**  We used random forests [43], an ensemble classification and regression-tree approach, to classify individual behavior throughout the study period based on the accelerometer output. Random forests have been utilized to classify accelerometer data from a variety of wildlife species with high accuracy (e.g., [44–47]). We selected 4 behaviors to classify, based on previous studies [47–51]. These behaviors included resting (individual is stationary, either laying or standing), headshaking (quick rotation of the head), locomotion (individual moves from one location to another, running or walking), and feeding (individual is consuming forage, pelleted feed or grass). Our main goal was to identify the amount of head-shaking that occurred as a means to assess changing levels of agitation or discomfort. Training data were generated from video recordings collected on three oryx for a related behavior study that occurred from 16 December 2015 to 18 January 2016 at the Fossil Rim Wildlife Center in Glen Rose, Texas, USA (Cunningham et al. In review). GPS collars were reused for this study.

High-impact events were used as landmarks to assist in synchronizing the collar data with the video clock. We identified 23,453 records, or 48.3 minutes, of the four behaviors in the video recordings. From these data, we randomly selected 5 minutes (2,400 records) each of locomotion, resting, and feeding and approximately one third of the 33 seconds of headshaking behavior, as these were rare events, to use for training (approximately 30% of the labeled dataset). We reserved the remainder of the identified behaviors for validation. Static acceleration (i.e., acceleration resulting from the position of the device in relation to the gravitational field) was extracted from the raw data using a 2-second moving window [51–53]. Results were then subtracted from the raw values to determine dynamic acceleration—the acceleration resulting from movement [51,53]. In addition to static and dynamic acceleration, we calculated the running minimum and maximum of each axis and the vectorial sum of overall body acceleration (VeDBA; [54]), to include as predictor variables in the random forests model.

Using performance metrics described by [50], we validated the random forests classification using a test dataset of approximately the same size as the training set and calculated the precision, recall, and accuracy for each behavior. Precision is the proportion of positive classifications that were correctly classified (i.e., measure of omission error), recall is the proportion of records that were correctly classified as true positive or true negative (i.e., commission error), and accuracy is a measure of the overall proportion of correctly assigned behaviors [50,55]. Following model validation, we used the fitted model from the three training individuals to classify the behavior of each individual in our current study based on their accelerometer data.

**Statistical analyses.**  We used a time series analysis to assess the duration of elevated head-shaking behavior identified by the random forests classification as a proxy for potential agitation or discomfort after collaring. Headshaking frequency was summarized on an hourly basis as a binomial count of classified headshaking records with a total sample size of 28,800 (generally 8 classified behaviors/second x 3,600 seconds/hour). We then compared three different models that represented headshaking frequency as a function of time since collaring (in hours). The best fitting model was identified by performing leave-one-out cross validation, selecting the model that minimized the sum of squares error.

Our first model was our baseline model for post-collaring headshaking behavior that assumed the hourly count of headshaking, $y_t$, was a binomial random variable with a time-dependent probability of headshaking, $p_t$, and the number of binomial trials, $N_t$, equal to the total number of accelerometer records across all behaviors ($N_t \approx 28,800$). We used a logistic link function to model the logit-scale probability of headshaking as an exponential decay process:

$$y_t \sim Binomial(p_t, N_t)$$

$$logit(p_t) = ae^{-bt} + c$$

where $a$ represents the initial change in headshaking due to collaring, $b$ represents the decay rate, and $c$ is an asymptote representing the baseline level of headshaking in the absence of a collaring effect.

Similarly, our second model also treated the frequency of headshaking, $y_t$, as a binomial response with the logit-scale probability of headshaking assumed to change over time based on an exponential decay process. However, we accounted for variation in the level of headshaking between day and nighttime periods, expected due to when animals are active and allowed to graze on pasture (8:00 to 16:00) and when animals are placed inside the barn facility are largely inactive (17:00 to 7:00), by including an offset term:

$$y_t \sim Binomial(p_t, N_t)$$

$$p_t = \begin{cases} \varphi_t, & if \ Day_t = 0 \\ \varphi_{t^d}, & if \ Day_t = 1 \end{cases}$$

$$logit(\varphi_t) = ae^{-bt} + c$$

where $Day_t$ is a binary indicator variable representing whether each count occurred during the daytime ($Day_t = 1$) or nighttime ($Day_t = 0$) and $d$ is an estimated exponent parameter that allows for a relative amplification (or reduction) in the probability of headshaking during the daytime.

Our final approach was to model the time series of hourly headshaking data as a combination of two harmonic processes that represented post-collaring and background levels of headshaking, respectively:

$$y_t \sim Binomial(p_t, N_t)$$

$$logit(p_t) = \sum_{k=1}^{2} \alpha_{kt}(\mu_k + U_{k1}cos(2\pi\omega t) + U_{k2}sin(2\pi\omega t))$$

$$\alpha_{1t} = e^{-bt} \quad \alpha_{2t} = 1 - \alpha_{1t}$$

The frequency of headshaking, $y_t$, was treated as a binomial response with a time-specific probability of headshaking, $p_t$, and known number of binomial trials, $N_t$. The probability of headshaking was then specified as a mixture of harmonic processes (e.g., [56]), with two complementary mixture weight parameters, $\alpha_{1t}$ and $\alpha_{2t}$, indicating the proportional contribution of each harmonic process, $k$, to the logit-scale probability of headshaking at each time $t$. The two harmonic processes were each composed of a process-specific mean, $\mu_k$, and a background series of daily oscillations, $U_{k1} cos(2\pi\omega t) + U_{k2} sin(2\pi\omega t)$, where $U_{k1}$ and $U_{k2}$ are coefficients

estimated from the data, and $\omega$ defines the number of cycles per unit time. In our study, was fixed at 1/24 because one hour represents 1/24th of an animal's daily activity cycle. Finally, the mixture weights for the first harmonic process, $\alpha_1$, were modelled as the outcome of an exponential decay process, $e^{-bt}$, where parameter $b$ defines the decay rate of the exponential function. Thus, the proportional contribution of the post-collaring harmonic process declines from 1 at the time of collaring ($t = 0$) towards an asymptote of 0, which corresponds to a complementary increase in the importance of the baseline process. Similarly, as the time since collaring increases, the contribution of the first harmonic process should diminish, such that the normal background rate can be estimated when the weight of this initial harmonic process equals 0.

Each of the three models for the accelerometer data incorporated the same exponential decay function to describe the transition (either increase or decrease) from post-collaring to pre-collaring levels of headshaking. This allowed us to calculate the predicted 'half-life' of the treatment effect in each model as $\ln(2/b)$, representing the time required for the treatment effect to decline to half its initial magnitude. Although the exponential process theoretically assumes headshaking never reaches its asymptote at baseline levels, we believe this model is a reasonable approximation for observed habituation since the expected difference from baseline after several half-lives is likely insignificant relative to background hourly variation. Further details are provided in S3 Code.

## Results

### Behavior

None of the observed changes in behaviors indicative of irritation supported our prediction of long-term adverse effects (i.e., Pre-treatment < Treatment = Post-treatment). Headshaking, however, increased significantly in frequency between the Pre-treatment (median: 0.05; CI: 0.04–0.07) and Treatment periods (median: 0.08; CI: 0.05–0.10), before declining Post-treatment to a level below the Pre-treatment baseline (median: 0.03; CI: 0.02–0.04) (Fig 2). This is consistent with a short-term effect that was detectable during the period 0–3 days post-collaring (i.e., Pre-treatment < Treatment > Post-treatment). Laying and scratching behaviors were also observed to significantly change from the Pre-treatment period. In both cases, however, the resulting behaviors declined in the Treatment and/or Post-treatment periods (Fig 2). These responses did not fit any of the expected patterns indicative of adverse effects from fitting GPS collars on oryx.

The majority of observations were comprised of behaviors not apparently related to adverse responses to collaring. Animals were most often observed with their heads raised, accounting for > 42% of all behaviors. Other behaviors, such as head down and locomotion, were also observed frequently (> 20% and > 14%, respectively). In our study, we accounted for the potential correlation of behavioral responses within 10-minute observation periods and found that many behaviors were strongly correlated. Specifically, we found that head up, headshaking, locomotion, and scratching often occurred in combination within observation periods (correlation coefficient: > 0.6; S1 Fig). These behaviors showed negative or weak positive correlations with head down or laying behaviors (correlation coefficient: < 0.3; S1 Fig). Headshaking most often occurred when animals' heads were raised (correlation coefficient: 0.82; S1 Fig). A full summary of predicted probabilities for all behaviors across treatment periods is provided in S1 Table.

### Fecal glucocorticoid metabolite concentrations

The best-fitting model for animals fitted with GPS collars was the Handling Response model (Fig 1). This model outperformed the other candidate models (provided in S1 Cross

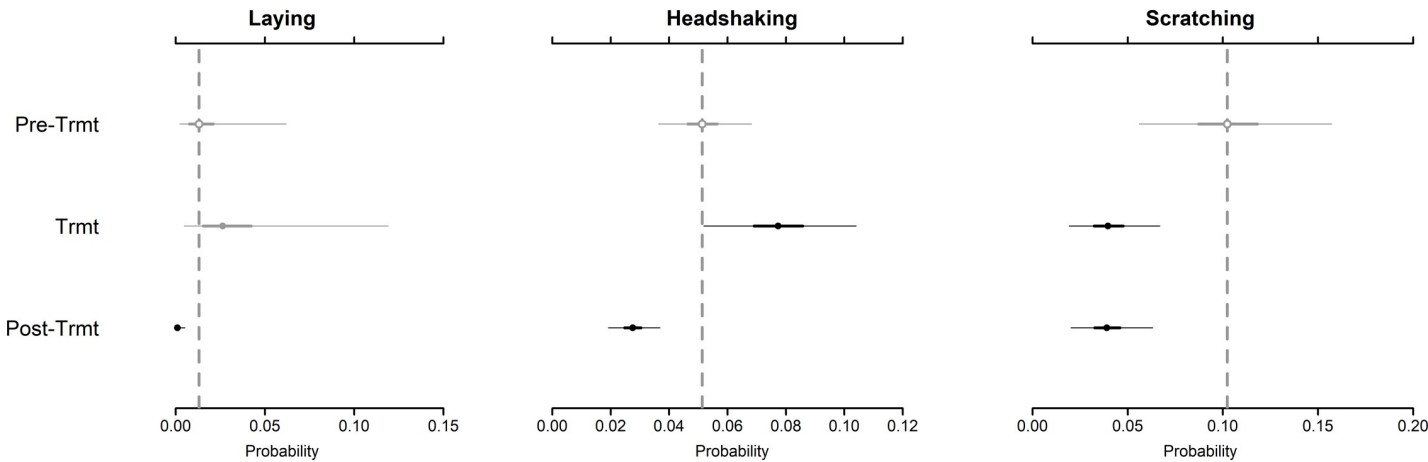

**Fig 2. Estimated changes in three behaviors indicative of adverse responses to GPS collaring by scimitar-horned oryx (Oryx dammah).** Predicted frequencies of each behavior are represented by point estimates (Bayesian posterior medians) and credible intervals (95% CIs—thin lines; 50% CIs—thick lines) across three treatment periods (Pre-Trmt: Pre-treatment; Trmt: Treatment; Post-Trmt: Post-treatment). Vertical dashed lines represent the posterior median for each parameter during the pre-treatment period (control). Credible intervals overlapping the pre-treatment median are colored gray (non-significant difference) or black (significant difference). Open circles indicate overlap with the 50% credible interval; closed circles indicate overlap with the 95% credible interval. Summary statistics for all parameters are provided in S1 Table.

Validation). Results of this model include a baseline intercept ($\beta_0$ mean: 43.35 ng/g; CI: 40.60–46.24 ng/g) with a +6.59 ng/g ($\beta_1$) increase in the intercept on the day after animals were fit with GPS collars ($k_1$), resulting in a FGM concentration of 49.94 ng/g (CI: 35.01–61.85 ng/g)

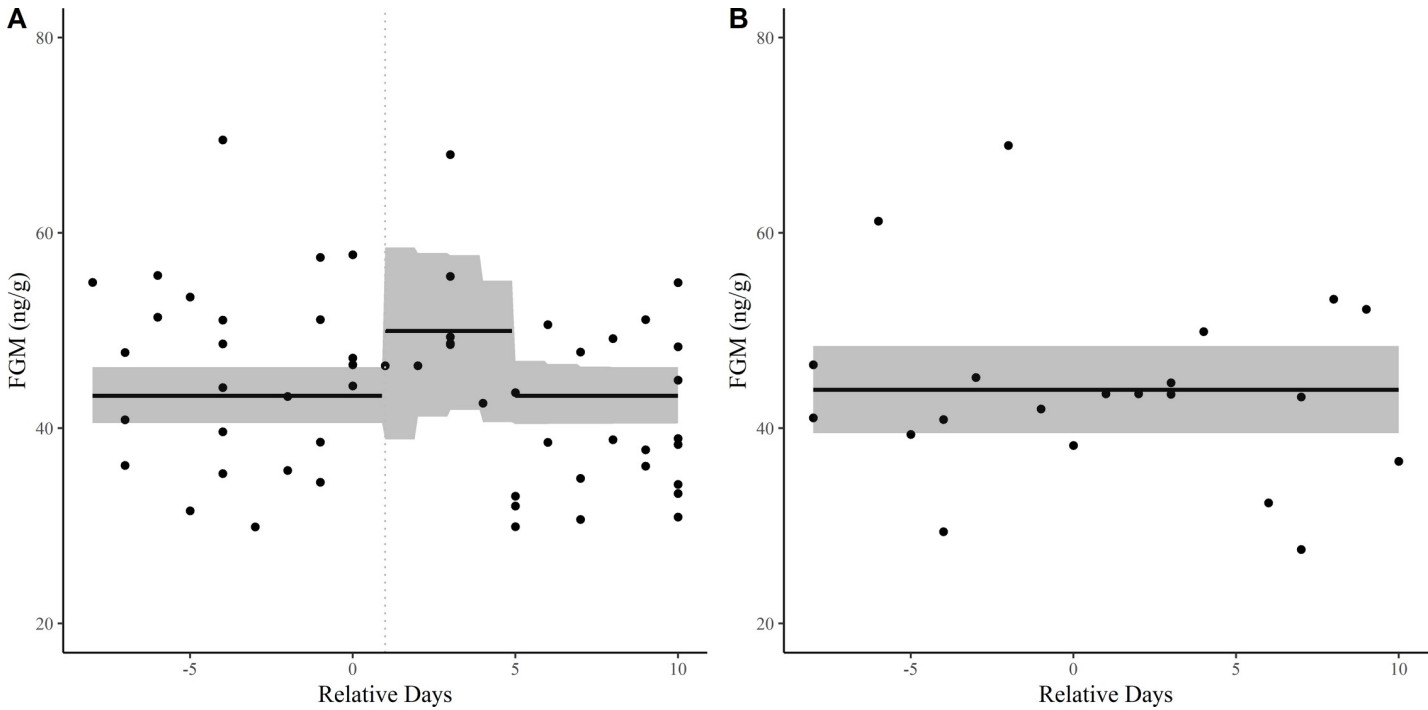

**Fig 3. Best-fit piecewise regression models predicting fecal glucocorticoid metabolites (FGM) in captive scimitar-horned oryx (Oryx dammah).** Treatment (A) and Control (B) animals displayed, representing animals fit with GPS collars (treatment) and those that were not (control). Treatment animals were observed to have a handling response, while Control animals exhibited no response. 95% posterior credible intervals represented by shaded area. Stress response is time lagged (Day 1—dashed vertical line).

during the treatment period. We estimated $k_2$, the day in which FGM concentrations returned to pre-treatment levels, as 5.03 days (CI: 2.36–9.81) from the date in which a rise in circulating FGMs was first detected (Fig 3). Estimated parameters and predicted responses from each model are included in S1 Cross Validation.

Control animals exhibited stable levels of FGMs across the study period (Fig 3), with no noticeable increase in metabolized hormones during animal handling activities. The 'No Response' model outperformed all other candidate models for control animals. Mean FGM concentration across the study period was 43.95 ng/g (CI: 39.42–48.43 ng/g). Parameter estimates from each of the models fit to control animals are included in S2 Cross Validation.

### Activity

Animals ($n$ = 4) were monitored with tri-axial accelerometers for 28–40 days, providing detailed information on the activity of each animal fitted with a GPS collar. Random forest models were used to classify animal behaviors (feeding, headshaking, resting, and locomotion) from the accelerometer data with a high level of accuracy (98.76% overall accuracy). Headshaking was classified with the highest accuracy (99.86%), but lowest precision (89.66%). This indicates that headshaking can clearly be identified when it occurs (i.e., headshaking has a distinct signature), but is slightly overclassified by our model. The recall rate for headshaking was 98.73%. Classification results for all behaviors are provided in S2 Table.

The harmonic model (i.e., binomial response treated as a mixture of harmonic processes) best predicted headshaking for three out of four animals fitted with Vectronic accelerometers. For the remaining animal, the day/night model (i.e., binomial response with a day/night switch) better fit the data (see S3 Cross Validation). Both models, however, captured the normal daily periodicity in headshaking, highlighting a consistent increase in headshaking during daytime hours (Fig 4).

Headshaking peaked in the hours immediately following GPS collar fitting, orders of magnitude higher than the estimated headshaking rate of a 'recovered' animal. Model dependent, GPS collar fitting (i.e., treatment effect) was estimated to cause a 480.54 (CI: 55.51–5068.14) to 742.10 (CI: 214.81–1197.93) percent increase in headshaking from the observed headshaking rate of a recovered individual (Table 3). Within 4 hours, however, the post-collaring increase in headshaking was estimated to have declined to half its initial magnitude (Table 3). The half-life of the treatment effect predicted by the day/night switching model was 3.16 hours (CI: 2.24–3.82) and 3.90 hours (CI: 1.70–5.36) by the harmonic model. Elevated levels of headshaking continued to be observed the day after collar fitting (Fig 4). Estimated parameters from each model and across each individual are provided in S3 Cross Validation.

### Discussion

Using a series of datasets and methodologies, our results demonstrate that captive scimitar-horned oryx experienced limited short-term effects after being fitted with GPS collars. Importantly, these effects subsided to pre-collaring levels within a few hours to a few days of collar fitting, most closely constituting a combination of handling (FGM analyses) and habituation (Accelerometer analyses) responses, with animals adapting to being physically restrained and adjusting to the device. Importantly, none of our analyses indicated long-term adverse effects that would have more serious animal welfare concerns. This is especially true when considering the significant benefit of using GPS collars to improve our understanding of scimitar-horned oryx ecology and inform conservation and management of the species upon reintroduction [57].

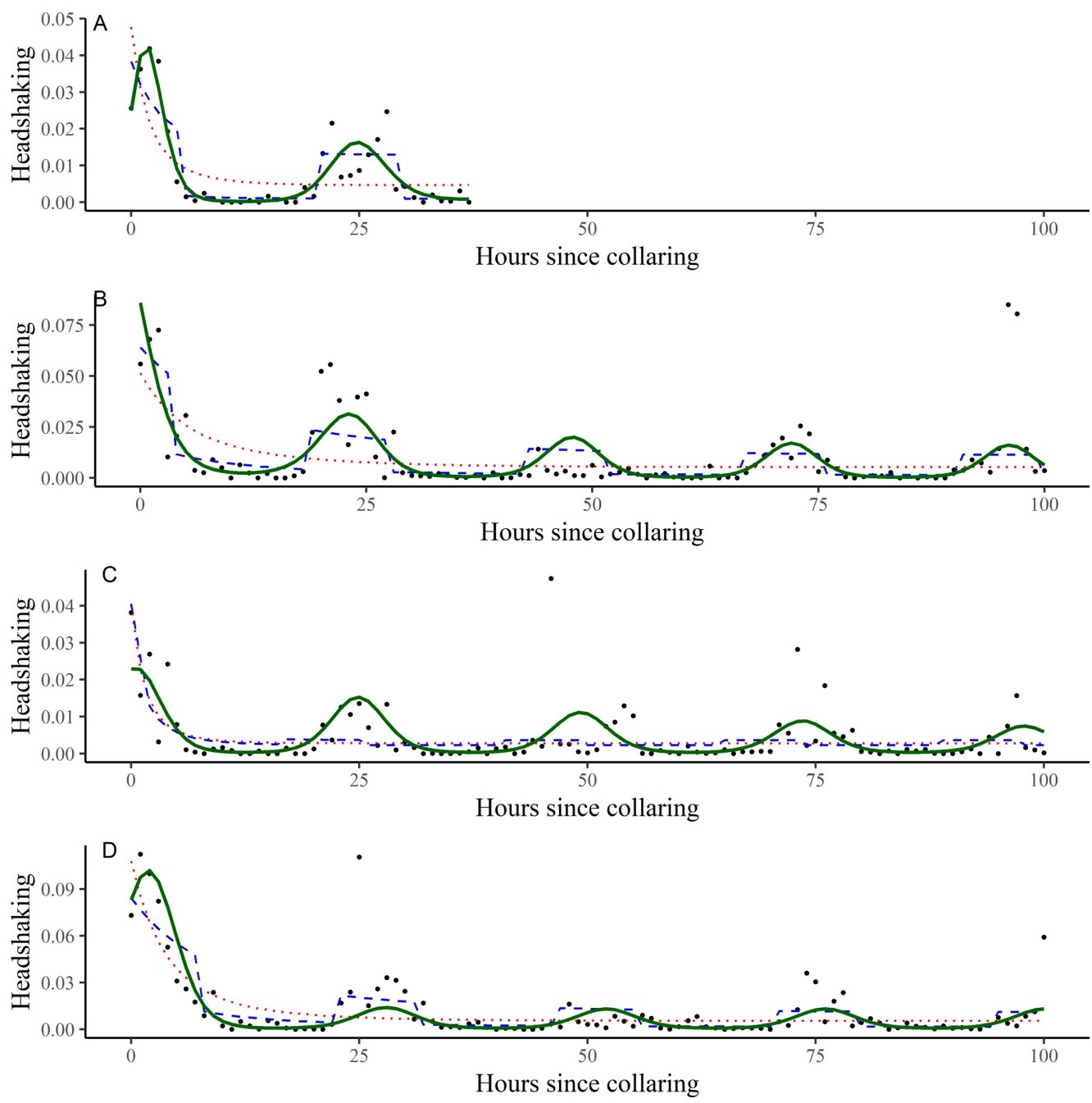

**Fig 4. Time-series models predicting headshaking activity in captive scimitar-horned oryx (Oryx dammah).** Each figure (A—114839, B—114426, C—114915, D—114969) represents a different animal (see Table 1). Headshaking was identified from a random forests classification of tri-axial accelerometer (8 Hz) data from Vectronic GPS collars and aggregated to an hourly count. Results show an increase in headshaking activity immediately after GPS collar fitting. Red dotted line—Binomial regression w/ negative exponential decay; Blue dashed line—Binomial regression w/ negative exponential decay + day/ night switch; Green solid line—Binomial regression w/ negative exponential decay + harmonic processes. Headshaking frequency shown as a frequency as black dots. Results truncated to 100 hours for visualization. See S3 Cross Validation for full time-series predictions and estimated parameters from each individual/model.

Accelerometry data showed that headshaking declined in magnitude by half every 4 hours after GPS collar fitting, pinpointing the time it takes for collaring effects to subside. In addition to the valuable information provided to wildlife researchers, these data should also assist in the

**Table 3. Estimated parameters from binomial regression models predicting headshaking in scimitar-horned oryx (*Oryx dammah*) after being fit with GPS collars.** Results summarize the joint posterior distributions across each individual (*n* = 4). Model 3 (Binomial regression with a negative exponential decay and a combination of harmonics) best fit the data in three out of four animals. Model 2 (Binomial regression with a negative exponential decay and a day/night switch) outcompeted the harmonic model in one of our animals. *Half-life* is the time (hours) required for the treatment effect (collar fitting) to decline to half its initial magnitude. Treatment Effect [(*Handling Rate–Recovery Rate)/Recovery Rate* * 100] is the percent increase in headshaking when comparing the rate of headshaking after being fit with GPS collars (*Handling Rate*) with the background rate (*Recovery Rate*). Model parameters and results from leave-one-out cross-validation provided in S3 Cross Validation.

| | μ | SD | median | 2.5% CI | 97.5% CI |
|---|---|---|---|---|---|
| Model 1: Binomial regression w/ exponential decay | 2.71 | 0.53 | 2.72 | 2.07 | 3.35 |
| Model 2: Binomial regression w/ negative exponential decay and day/night switch | 3.06 | 0.72 | 3.16 | 2.24 | 3.82 |
| Model 3: Binomial regression w/ negative exponential decay and a combination of harmonics | 3.21 | 1.17 | 3.41 | 1.48 | 5.30 |
| | μ | SD | median | 2.5% CI | 97.5% CI |
| Model 1: Binomial regression w/ exponential decay | 1286.97 | 427.25 | 1136.90 | 870.44 | 1988.42 |
| Model 2: Binomial regression w/ negative exponential decay and day/night switch | 721.09 | 345.19 | 747.25 | 215.03 | 1197.80 |
| Model 3: Binomial regression w/ negative exponential decay and a combination of harmonics | 2191.20 | 3170.55 | 684.14 | 55.39 | 9638.71 |

ongoing discussion with the general public, providing quantitative information on the time it takes for animals to adjust to being fitted with tracking devices (see also [13–17,39]). Although our sample sizes are small, our results were consistent across individuals monitored and are the first to quantify the time it takes for headshaking to return to normal levels after GPS collaring fitting. Accelerometers, however, are unable to provide pre-collaring data, underlying the importance of alternative forms of evidence. We estimated the 'background' rate by evaluating the contribution of the harmonic models at capturing the daily periodicity in headshaking, assumed to be a close analogue to a pre-collaring baseline.

Behavioral ethograms have been widely used to assess subtle changes in behavior over time (e.g., [35,58]). In our study, behavior observations were collected across all treatment periods, demonstrating that headshaking increased significantly after collars were fitted and decreased below pre-treatment levels during the post-treatment period. This result aligned directly with findings from our other two methodologies, even though the length of the treatment period was pre-defined, rather than estimated, in our analyses. The decline in headshaking during the post-treatment period could indicate a continued negative response to the collar, with animals shaking their heads less to limit collar movement and potential irritation. Locomotive behavior (walking/running), however, remained unchanged throughout the entire study period, as did the position of the head of animals (head up or head down) while standing/feeding, an initial concern of partners involved in the reintroduction. Other studies (e.g., [37]) have observed that collars can impede normal feeding behavior, a situation to be avoided.

Glucocorticoid analyses provided valuable information on baseline conditions that would otherwise be difficult to obtain. Successfully incorporated to investigate the effects of radio-collaring on African wild dog (*Lycaon pictus*, [59]), this non-invasive technique has been broadly applied to address the effects of environmental stressors on wildlife (e.g., [60–62]). During the treatment period and in combination with the increase in FGM concentrations that we observed, variability across animals decreased. This is of particular note considering the large degree of variability that commonly results because of age, sex, reproductive status, and social status in FGM analyses [63–67], factors unaccounted for in our analyses. Admittedly, collecting fecal samples from free-ranging populations requires substantial effort. However, if collected from known individuals, as in [61] and [68], FGM analyses offer a non-invasive means to evaluate the physiological state of animals over time that could be of great benefit to the monitoring of populations during reintroduction efforts.

Although negative effects from collars appeared to be limited in our study, there may be other indirect effects that may not be apparent for some time after devices have been fit on

animals. Device-induced drag on migrating seabirds, for example, may result in increased energy expenditure and result in a loss in body condition and/or increased mortality risk [69,70]. In the case of reintroduced oryx, the burden of the GPS collar could be a concern if it incurs an additional energy cost to the animal, especially as animals acclimate to their new surroundings. Rasiulis et al. [71] suggest that even a small additional weight of a tracking device may adversely affect individuals that may already be experiencing declines in body condition, as would be expected upon release [72]. In our study, collars represented < 2% of the animals' body weight, well below general guidelines [11], but still posing risks that should be considered.

Secondary effects of a lowered body condition could also include reduced rates of reproduction [73–75], which would preclude the establishment of self-sustaining populations. Five of eight oryx (63%) included in our study declined in body weight over the study period, changes that were within a normal range of variation for each animal. While multiple factors could explain these weight changes (environmental conditions, individual variation, parasites), we recommend that whenever possible, researchers use standardized indices (e.g., [76]) to track the body condition of tracked individuals over long time periods and collect other metrics (e.g., changes in social hierarchies/interactions, reproductive status) that could offer insight into the effects resulting from tracking devices [39].

We recognize that most researchers are unable (both financially and logistically) to study the effects on animals once they are fit with GPS collars. Zoos, however, offer one of the few possibilities to address research questions that have the potential for direct benefit to wild populations, even if conditions do not match those of wild populations. In addition to providing insight into the potential impact of tracking devices (as in [59]), captive studies also allow for experimental control, a scenario typically impossible for most studies [36]. We encourage researchers and manufacturers to collaborate with the zoo community and appropriately test devices prior to deployment. This could identify potential problems prior to deployment, saving thousands of dollars in tagging and travel costs, and further our understanding of species-specific effects.

We worked directly with each manufacturer and were able to provide direct feedback to improve the fit of each device prior to reintroduction efforts. In six of ten animals included in this study, we noted mild to moderate rubbing on their mandibles and/or neck ridgelines, symptoms common to other studies (e.g., [20]). Two animals also had very minor wear (broken guard hairs or small spots of hair loss) along their necks. Determining a good fit is vital. Even if rubbing does not result in damage to the skin itself in relatively dry environments (as observed in Cunningham et al. In Review), regions experiencing higher precipitation may engender the growth of microorganisms between the neck and collar, potentially increasing the risk of infection if the skin were to be ruptured by wear from the collar or from scratching by the animal [21,77]. These symptoms, caused by biting flies in two reintroduced oryx at our study site in Chad in 2018, were exacerbated by the collar. In these cases, these effects were so severe that collars were remotely removed via a drop-off mechanism to reduce infection and allow the skin to heal.

As suggested by Moll et al. [78], the combination of multiple forms of evidence, as included here, allow researchers to obtain a broader view of the effects of marking individuals. For species included in reintroduction programs, where the stresses to each individual are high, it is paramount to understand the effects of devices aimed to monitor individuals before release. Our study illustrates that effects from GPS collar fitting were short-term and provided no foundation for welfare concerns. Due to the enduring question of if and how monitoring devices impact wildlife [39,77,79], we highly encourage researchers involved with wildlife tracking programs to evaluate the effects that monitoring devices have on the survival and fitness of the individuals they aim to save.

## Conclusions

GPS collars have proven essential elements to an ecologist's toolbox, vital for assessing conservation actions, and one of the only cost-effective means for evaluating the fate of every individual in reintroduction efforts, especially across large remote areas. Tracking devices, however, also have potential risks and may burden their carriers with additional stress, causing injury or even death [77]. Our study found no such animal welfare concerns from GPS collars fitted on captive scimitar-horned oryx. While we are unable to disentangle the effects of animal handling from the effects related solely to GPS collars themselves, we illustrate that significant effects in behavior, activity, and FGMs dissipate quickly after animals are fitted with a GPS collar. We encourage further investigation into the long-term effects of these revolutionary devices to the field of movement ecology, but concur with similar studies ([14–17,39,80]) that find no evidence that should preclude their use.

## Supporting information

**S1 Code. Behavioral data and analysis.** Script and data tutorial to investigate behavioral changes observed in scimitar-horned oryx (*Oryx dammah*) fit with GPS collars. Data fit in a Bayesian framework, estimating the probability of each behavioral activity and based on a multinomial likelihood. Each animal was used as their own control to assess how each behavior changed across time periods (Pre-treatment, Treatment, Post-treatment).
(ZIP)

**S2 Code. Fecal glucocorticoid metabolite data and analysis.** Script and data tutorial to investigate changes in fecal glucocorticoid metabolite levels in scimitar-horned oryx (*Oryx dammah*). Data fit in a Bayesian framework, testing various piecewise regression models to predict the response of animals fitted with GPS collars. Data split between treatment (collared) and control (non-collared) animals.
(ZIP)

**S3 Code. Accelerometer data and analysis.** Script and data tutorial to quantify the short-term decline in headshaking that occurred after scimitar-horned oryx (*Oryx dammah*) were fitted with GPS collars. Headshaking, captured in data collected from tri-axial accelerometers, was classified by random forest models with a high degree of accuracy and precision. Data were then aggregated to an hourly time window. Competing time-series models were then used to calculate the amount of headshaking that occurred over time.
(ZIP)

**S1 Table. Summary of animal behaviors across treatment periods.** Parameter estimates from a multinomial regression model predicting animal behavior before (Pre-Treatment), during (Treatment), and after (Post-Treatment) periods for scimitar-horned oryx (n = 10) fit with GPS collars. Posterior mean, median, standard deviation (SD), and 95% credible interval (CI) provided. See Table 2 for behavior category descriptions.
(DOCX)

**S2 Table. Random forests classification results.** Classification accuracy, precision and recall of behaviors identified by a random forest model in the analysis of accelerometry data (8 Hz) recorded in Vectronic GPS collars fit on four (n = 4) scimitar-horned oryx (Oryx dammah).
(DOCX)

**S1 Fig. Behavior correlation matrix.** Pearson's correlation coefficients of animal behaviors identified within observation windows for captive scimitar-horned oryx (*Oryx dammah*). Head Up (HU), Headshaking (HDSK), Locomotion (LOCO), and Scratching (SCRATCH)

most often occurred in combination with one another. Head Down (HD) and Laying (LAY) infrequently occurred in combination with these behaviors.
(DOCX)

**S1 Cross Validation. Fecal glucocorticoid metabolite validation results–Treatment animals.** Leave-one-out cross-validation results from each of four (4) piecewise regression models, evaluating fecal glucocorticoid metabolites ($n_{obs}$ = 55) collected from scimitar-horned oryx (*Oryx dammah*) fit with GPS collars (Treatment). Models fit in a Bayesian framework and evaluated by summing the squared errors (SSE). The Handling Response model was identified as the best model. We ran three parallel Markov chain Monte Carlo (MCMC) chains for 400,000 iterations, discarding the first 80,000 iterations (20%) of each chain as burn-in, and thinned the remaining posterior samples (1:100) from the joint posterior distribution for each model.
(DOCX)

**S2 Cross Validation. Fecal glucocorticoid metabolite validation results–Control animals.** Leave-one-out cross-validation results from each of four (4) piecewise regression models, evaluating fecal glucocorticoid metabolites ($n_{obs}$ = 21) collected from scimitar-horned oryx (*Oryx dammah*)–control animals (not fit with GPS collars). Models fit in a Bayesian framework and evaluated by summing the squared errors (SSE). The No Response model was identified as the best model. That is, this model minimized the Sum of Squared Error (noted in bold). We ran three parallel Markov chain Monte Carlo (MCMC) chains for 400,000 iterations, discarding the first 80,000 iterations (20%) of each chain as burn-in, and thinned the remaining posterior samples (1:100) from the joint posterior distribution for each model.
(DOCX)

**S3 Cross Validation. Validation results for time-series models.** Leave-one-out cross-validation results evaluating three (3) time-series models fit to predict headshaking behavior in scimitar-horned oryx (*Oryx dammah*). Data derived from tri-axial accelerometers fit on four ($n$ = 4) animals, recording eight activities per second (8 Hz). Headshaking was identified via random forest, an ensemble classification and regression tree. Results were aggregated to an hourly interval. Models predicting the number of hourly headshakes fit in a Bayesian framework and evaluated by summing the squared errors (SSE). The Harmonic model was identified as the best model (noted in bold) in three out of four cases. We ran three parallel Markov chain Monte Carlo (MCMC) chains for 400,000 iterations, discarding the first 80,000 iterations (20%) of each chain as burn-in, and thinned the remaining posterior samples (1:100) from the joint posterior distribution for each model. Predicted responses and estimated parameters from the joint posterior distributions are provided for each animal/model.
(DOCX)

## Acknowledgments

We thank J.Kolowski and four anonymous reviewers for constructive comments that greatly improved the quality of this manuscript, the staff at the Fossil Rim Wildlife Center, C. Kochanny at Vectronic Aerospace for fielding our countless questions, and the continued professionalism of the husbandry and veterinary staff at the Smithsonian Conservation Biology Institute for assistance with immobilization procedures.

## Author Contributions

**Conceptualization:** Jared A. Stabach, Melissa Songer, Tim Wacher, Steven Monfort, Peter Leimgruber.

**Data curation:** Jared A. Stabach, Stephanie A. Cunningham, Joel L. Mota.

**Formal analysis:** Jared A. Stabach, Stephanie A. Cunningham, Grant Connette.

**Funding acquisition:** Jared A. Stabach, Melissa Songer, Pierre Comizzoli, John Newby, Steven Monfort, Peter Leimgruber.

**Investigation:** Jared A. Stabach, Stephanie A. Cunningham, Joel L. Mota, Dolores Reed, Michael Byron.

**Methodology:** Jared A. Stabach, Stephanie A. Cunningham, Grant Connette.

**Project administration:** Jared A. Stabach, Melissa Songer.

**Resources:** Jared A. Stabach.

**Software:** Jared A. Stabach, Stephanie A. Cunningham, Grant Connette.

**Supervision:** Jared A. Stabach, Dolores Reed, Melissa Songer, Janine L. Brown.

**Validation:** Jared A. Stabach, Grant Connette, Michael Byron.

**Visualization:** Jared A. Stabach.

**Writing – original draft:** Jared A. Stabach, Stephanie A. Cunningham, Grant Connette, Joel L. Mota, Michael Byron.

**Writing – review & editing:** Jared A. Stabach, Stephanie A. Cunningham, Grant Connette, Joel L. Mota, Dolores Reed, Michael Byron, Melissa Songer, Tim Wacher, Katherine Mertes, Janine L. Brown, Pierre Comizzoli, John Newby, Steven Monfort, Peter Leimgruber.

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
