## [Decision Letter · Decision Letter 0]

5 Nov 2019

PONE-D-19-23204

Short-term effects of GPS collars on the activity, behavior, and adrenal response of scimitar-horned oryx (Oryx dammah)

PLOS ONE

Dear Dr. Stabach,

Thank you for submitting your manuscript to PLOS ONE. After careful consideration, we feel that it has merit but does not fully meet PLOS ONE’s publication criteria as it currently stands. Therefore, we invite you to submit a revised version of the manuscript that addresses the points raised during the review process.

We would appreciate receiving your revised manuscript by Dec 20 2019 11:59PM. When you are ready to submit your revision, log on to https://www.editorialmanager.com/pone/ and select the ‘Submissions Needing Revision’ folder to locate your manuscript file.

To enhance the reproducibility of your results, we recommend that if applicable you deposit your laboratory protocols in protocols.io, where a protocol can be assigned its own identifier (DOI) such that it can be cited independently in the future. For instructions see: http://journals.plos.org/plosone/s/submission-guidelines#loc-laboratory-protocols

A rebuttal letter that responds to each point raised by the academic editor and reviewer(s). This letter should be uploaded as separate file and labeled ‘Response to Reviewers’.A marked-up copy of your manuscript that highlights changes made to the original version. This file should be uploaded as separate file and labeled ‘Revised Manuscript with Track Changes’.An unmarked version of your revised paper without tracked changes. This file should be uploaded as separate file and labeled ‘Manuscript’.

We look forward to receiving your revised manuscript.

Kind regards,

Edna Hillmann, Ph.D.

Academic Editor

PLOS ONE

Journal Requirements:

Please ensure that your manuscript meets PLOS ONE’s style requirements, including those for file naming. The PLOS ONE style templates can be found at

Additional Editor Comments (if provided):

Since it was not possible to find a second reviewer for this manuscript, I asked the journal office if it is possible to review the manuscript as academic editor. Thus, I would like to provide my own recommendation together with reviewer 1, and I ask you to consider our recommendations (or argue against them).

Academic Editor´s Review:

Study area

Are the conditions of the study area in any case transferable to the African habitat the Oryx would be reintroduced? How do you expect predator presence to affect potential stress responses to GPS-collars? This, of course, is rather speculative but seems to be relevant for application GPS collars for wild animals.

Table 1 and Discussion

There was a weight loss found in 5 of 8 animals (as stated in the discussion). Can you exclude an influence of the treatment? This potential welfare risk should be discussed in more detail.

Line 159: It is not clear how long the pre-treatment and post-treatment period were. Please give more details. I assume these periods also lasted for three days? If not, please justify.

Lines 184ff: The behavioural sampling (direct observation) seems to be very short. 10 min/day? Please justify this short observation time, and if this short duration properly reflects the behaviours you were interested in. The total number of observation periods is not so relevant, more important is the observation time per animal per treatment as given in parentheses.

Line 205ff: I strongly recommend shortening the Statistic Chapters. Is it necessary to give all formulas in detail? Some of the equations resemble “standard regression”, no? If so, this section should be condensed.

Lines 267ff: I suggest deleting Fig.1 since all information is given in the text.

Line 414: There is a word missing after “In our study, XXX was fixed…”

Figure 3: I do not understand the use of this correlation matrix. How does it help to answer the research question?

Line503ff: Headshaking was analysed for both direct observation and accelerometer data. Except for validation purposes, I recommend deleting headshaking from the analysis of direct observations.

Line553: If emphasizing the application of Bayesian statistics is important, why did you choose this approach? Either explain or leave the sentence out.

Line 565: weight loss would be a welfare concern

Reviewers’ comments:

Reviewer’s Responses to Questions

**Comments to the Author**

1. Is the manuscript technically sound, and do the data support the conclusions?

Reviewer #1: Partly

2. Has the statistical analysis been performed appropriately and rigorously? 

Reviewer #1: I Don’t Know

3. Have the authors made all data underlying the findings in their manuscript fully available?

Reviewer #1: Yes

4. Is the manuscript presented in an intelligible fashion and written in standard English?

Reviewer #1: Yes

5. Review Comments to the Author

Reviewer #1: Line 38: Name the two specific collars used.

Line 40: Fecal corticosterone metabolites in place of stress hormones

Line 123: Were any of the animals subjected to collars of any type prior to the research?

Line 129: Would it be a better design to use the same animals as control animals whereby the

animals are provided treatment and then measurements are done without treatment.

Line 137: Did capture require any chemical method such as tranquilizer? Please provide details.

Line 138: Please provide photo of animal within the TAMER

Line 139: Could the procedure of AI have impacted FGM readings?

Line 145: Semen collection is a standard procedure which may generate physiological stress in wildlife. It is unclear if the experimental design for testing the effects of GPS collaring is correct if other procedures were done in combination and it makes it difficult to tease apart the cause-effects?

Line 175: please verify how the behaviour aspect presented in the current manuscript is different from the recent publication (BioRxiv)?

Line 251: When were the assays performed since sample collection in 2015?

Line 255: It is unclear why corticosterone instead of cortisol (a major glucocorticoid of mammals) was measured?

Line 257: Please provide the sensitivity of the RIA.

Line 267: Is this hypothesis correct if animals were sampled for semen and females were AI’ed during collaring?

Line 282: As the gut-passage time been biologically validated in the study species? Please provide a reference for the gut-passage time

6. PLOS authors have the option to publish the peer review history of their article (what does this mean?). If published, this will include your full peer review and any attached files.

Reviewer #1: No

---

## [Author Response · Author response to Decision Letter 0]

23 Dec 2019

Manuscript Number: PONE-D-19-23204

Title: Short-term effects of GPS collars on the activity, behavior, and adrenal response of scimitar-horned oryx (Oryx dammah)

Authors: JA Stabach, SA Cunningham, G Connette, JL Mota, D Reed, M Byron, M Songer, T Wacher, K Mertes, JL Brown, P Comizzoli, J Newby, S Monfort, P Leimgruber

Responses to Reviewer Comments

Academic Editor:

Since it was not possible to find a second reviewer for this manuscript, I asked the journal office if it is possible to review the manuscript as academic editor. Thus, I would like to provide my own recommendation together with reviewer 1, and I ask you to consider our recommendations (or argue against them).

 Response: We thank both the academic editor and independent reviewer for their insightful and constructive comments which have greatly improved the quality of this manuscript. Itemized responses to each question raised are provided. We feel the main critique of the manuscript was the length of our methods section, which provided detailed descriptions of eight separate models. As a result, we have significantly shortened these sections, where possible. This was difficult for the time series analyses, because the models provided are custom non-linear functions. We have also clarified the complexities of our study and emphasized the difficulties associated with disentangling GPS collaring effects from capture effects. 

Academic Editor Comments:

Study area

Are the conditions of the study area in any case transferable to the African habitat the Oryx would be reintroduced? 

Response: No. Habitats are different between our study area (temperate, managed pasture) and our reintroduction site (Sahelian grassland). We, however, do not aim to make comparisons of survivability across these different systems. Instead, we focus only on short-term effects that potentially occur from GPS collar fitting, with inference towards the length of time it takes for observed effects to subside. We have been cautious to not overstate the inference from our study, providing comments in our Discussion which highlight our limited sample size (page 29, line 601), draw attention to the fact that “conditions do not match those of wild populations (page 31, line 655), and note that effects to oryx “may not be apparent for some time after devices have been fit on animals” (page 30, line 634). Throughout our Discussion we urge further research into this topic, with a focus on longer term assessments. 

How do you expect predator presence to affect potential stress responses to GPS-collars? This, of course, is rather speculative but seems to be relevant for application GPS collars for wild animals.

Response: This is an important question and one we (and others) are keenly interested in. To our knowledge, however, no study exists to address this issue. Most researchers do try to match the color of the belting with the coat pattern of the species of interest, but it is unclear if this actually helps or hinders an animals’ ability to avoid predators. We feel this to be a very valid question, but one that we do not have data to answer. 

Table 1 and Discussion:

There was a weight loss found in 5 of 8 animals (as stated in the discussion). Can you exclude an influence of the treatment? This potential welfare risk should be discussed in more detail.

Response: Good point. From speaking with our animal husbandry team, the observed changes in weight are not out of the ordinary and could be the result of many factors (environmental conditions, individual variation, social factors, parasites). Important is that the changes observed over the short time period of our study did not signal any concerns amongst the team, even animal (114969) that is noted with a 11.8 kg decline in weight over the study period. 

In addition, we performed a sign test (Kitchens, 2003) on the observed sample of changes in body weight. We were unable to reject (p = 0.2188) the null hypothesis that a new observation would be equally likely to be positive or negative. Given our limited sample, this result does not necessarily mean collaring did not affect body weight, but does indicate that weight loss in 5 of 8 individuals does not provide sufficient evidence to conclude that there was an effect.

We have edited the text slightly in the Discussion (Page 31, Line 649), which focuses on the importance of collecting information on body condition over longer periods to evaluate potential long-term consequences of GPS collars.

Page 31, Line 645: Five of eight oryx (63%) included in our study declined in body weight over the study period, changes that were within a normal range of variation for each animal. While multiple factors could explain these weight changes (environmental conditions, individual variation, parasites), we recommend that whenever possible, researchers use standardized indices (e.g., [75]) to track the body condition of tracked individuals over long time periods and collect other metrics (e.g., changes in social hierarchies/interactions, reproductive status) that could offer insight into the effects resulting from tracking devices [38]. 

Reference for sign test:

Kitchens, L J.(2003). Basic Statistics and Data Analysis. Duxbury.

Line 159: It is not clear how long the pre-treatment and post-treatment period were. Please give more details. I assume these periods also lasted for three days? If not, please justify.

Response: We agree this is confusing, as there are multiple monitoring periods for each of the separate analyses, with difficulties associated with collecting samples from each individual on every occasion. We have updated Table 1, editing the column named ‘Monitoring Period (Days)’ to ‘GPS Monitoring Period (Days)’, better conveying that this period relates only to the time period in which animals were fitted with GPS collars. We also removed the days listed for animals that served as controls for the FGM analyses (animal 113204, 114427, and 114968), since these animals were not fitted with collars. On page 10 (line 191), we updated the number of observations within each treatment period to convey the number of days that animals were monitored. The average number of observations per individual, with standard deviations, provides an assessment of sample size per observation periods, and underlies important items in our analyses which account for repetitive measures, random effects, and unequal samples sizes across treatment periods (Now noted on page 12, line 224, and included in Appendix S1). 

Page 10, Line 197: Behavioral data were collected for a total of 198 observation periods (Pre-treatment (17 days) = 55 periods [5.5 ± 3.1 per individual]; Treatment (4 days) = 30 periods [3.0 ± 0.8 per individual]; Post-treatment (22 days) = 113 periods [11.3 ± 4.7 per individual]). ]). 

Lines 184ff: The behavioural sampling (direct observation) seems to be very short. 10 min/day? Please justify this short observation time, and if this short duration properly reflects the behaviours you were interested in. The total number of observation periods is not so relevant, more important is the observation time per animal per treatment as given in parentheses.

Response: We have now edited this section extensively, clarifying the number of behavioral periods, the number of behavioral observations per individual, and the total number of behaviors we observed. Despite the short duration of each observation period, inference about treatment effects was based on roughly 8,000 total behaviors observed from almost 200 observation periods. The sampling methodology used is a standard, validated focal sampling technique (i.e., Altmann, 1974) to identify changes in behavioral over time and has been widely used over the past few decades. We chose a simplified set of behaviors to monitor (6 in total), which mirror behaviors derived from Packard et al., (2014), are consistent with studies investigating collaring effects (i.e., Durnin et al., 2004; Nussberger & Ingold, 2006), and fit with our research objectives (i.e., Is the animal feeding more or less during the treatment period? Is the animal spending more time lying down? Has locomotion been curtailed? Does the animal tend to eat more or less with their head up or down? Do we see increases in potential irritation behaviors?). All of these questions can be addressed using the methodology we adopted. 

Altmann J. (1974) Observational study of behavior: sampling methods. Behaviour, 49, 227–67. 

Durnin M.E., Swaisgood R.R., Czekala N., & Hemin Z. (2004) Effects of radiocollars on giant panda stress-related behavior and hormones. Journal of Wildlife Management, 68, 987–992. 

Nussberger B. & Ingold P. (2006) Effects of radio-collars on behaviour of alpine chamois Rupicapra rupicapra rupicapra. Wildlife Biology, 12, 339–343. 

Packard J.M., Loonam K.E., Arkenberg C.R., Boostrom H.M., Cloutier T.L., Enriquez E.J., Eyres A., Haefele H., Salzar T.R., Smultea M.A., & Snodgrass K. (2014) Behavioral profiles of Africa bovids (Hippotraginae). Journal of Zoo and Aquarium Research, 2, 83–87. 

This section now reads:

Page 10, Line 197: Behavioral data were collected for a total of 198 observation periods (Pre-treatment (17 days) = 55 periods [5.5 ± 3.1 per individual]; Treatment (4 days) = 30 periods [3.0 ± 0.8 per individual]; Post-treatment (22 days) = 113 periods [11.3 ± 4.7 per individual]). In total, we observed 2215 behaviors during the Pre-treatment period (221.5 ± 124.4 per individual), 1211 behaviors during the Treatment period (121.1 ± 31.6 per individual), and 4544 behaviors during the Post-treatment period (454.4 ± 188.1 per individual). 

Line 205ff: I strongly recommend shortening the Statistic Chapters. Is it necessary to give all formulas in detail? Some of the equations resemble “standard regression”, no? If so, this section should be condensed.

Response: We have shortened our statistical chapters significantly, referring the reader to the appendices for further details. The time-series analyses, however, represent custom non-linear models that can’t be described by text alone. As a result, we have not changed this section.

Lines 267ff: I suggest deleting Fig.1 since all information is given in the text.

Response: We recognize the reviewer’s concern over duplicating efforts. However, from our experience, we have found it invaluable to provide a figure to explicitly describe the modeling outcomes, even when those same outcomes have been described in words. This is especially true given the international audience of PlosOne and the range in analytical skills of its readership. If not amenable to the reviewer, we will move this figure to an appendix, but feel that it provides a natural progression and further guides the reader to better understand our results, minimizing confusion of the modeling outcomes that were tested. 

Line 414: There is a word missing after “In our study, XXX was fixed…”

Response: Apologies. The number of cycles per unit time, referenced as the parameter ‘’’, was inadvertently deleted from the manuscript. The text has now been updated. 

Figure 3: I do not understand the use of this correlation matrix. How does it help to answer the research question?

Response: Agree. We have moved the correlation matrix to Appendix S4, as it is not central to the focus of the paper. The correlation matrix was simply meant to provide information to the reader and disentangle the behaviors that frequently occurred together. This was included as it was meant to remove potential concerns that certain behaviors would be expected to be found together.

Line503ff: Headshaking was analysed for both direct observation and accelerometer data. Except for validation purposes, I recommend deleting headshaking from the analysis of direct observations.

Response: We feel it is important to include both methods for assessing headshaking because they provide unique insights. Accelerometers provide continuous, around-the-clock information from the point of collaring onwards but behaviors are not known with 100% certainty. Unlike accelerometers, which must be fitted on animals, behavioral ethograms allow for comparison with pre-treatment baselines and behaviors are assigned with 100% certainty. These points are discussed in detail on page 29 of the Discussion (paragraphs 2 and 3).

Line 553: If emphasizing the application of Bayesian statistics is important, why did you choose this approach? Either explain or leave the sentence out.

Response: We have updated this sentence slightly, highlighting the use of a “series of datasets and methodologies” to evaluate short-term effects of GPS collars instead of focusing on the “non-standard regression models”. The application of Bayesian methods was ultimately a practical decision. While all of these analyses could have been formulated in a frequentist perspective, the final model (i.e., the harmonic model of the frequency of headshaking) would have been particularly difficult to do so. During earlier versions of this manuscript, we had formulated the piecewise regressions to evaluate FGM concentrations using frequentist methods. As expected, we found no difference in the inference between modeling methodologies. But, we decided that it made more sense for manuscript consistency to standardize the methodology used, which is why we used Bayesian methods for every model described. Both this section in the Discussion and Abstract have been updated.

Page 28, Line 586: Using a series of datasets and methodologies, our results demonstrate that captive scimitar-horned oryx experienced limited short-term effects after being fitted with GPS collars. 

Line 565: weight loss would be a welfare concern

Response: Addressed above with comment made about weights provided in Table 1 and noted in the Discussion.

 

Reviewer #1: 

Line 38: Name the two specific collars used.

Response: Details now provided.

Line 40: Fecal corticosterone metabolites in place of stress hormones

Response: Stress hormones removed.

Line 123: Were any of the animals subjected to collars of any type prior to the research?

Response: No. Information added to page 6, line 125.

Page 6, Line 125: No animal had ever before been fitted with a GPS collar.

Line 129: Would it be a better design to use the same animals as control animals whereby the

animals are provided treatment and then measurements are done without treatment.

Response: This was now been clarified. For Behavioral Observations, we observed all animals in each treatment period and accounted for repeated measurements of the same individuals using a random effect. For the Fecal Glucocorticoid Analyses, we created a control group (n=2) to separate those animals that received a collar and those that did not. Sentence updated slightly (Page 6, Line 126). These data are displayed in Table. We also state in our methods (Page 18, Line 350) that “We fit separate models were for control and treatment animals, providing a comparison with animals that were not fitted with a GPS collar but may be affected by disturbances caused during collaring related activities”. Even if we were able to isolate these control animals from the group during study activities, it could then be argued that changes to the structure of this group could then be attributed to observed changes in FGM levels. Thus, it is very difficult to differentiate these factors given logistical limitations and inherent sensitivities of the group.

Page 6, Line 126: Control animals (n=3), used as a comparison dataset for analyses of FGMs, were 7 to 18 years old.

Line 137: Did capture require any chemical method such as tranquilizer? Please provide details.

Response: Females were physically restrained without any chemical tranquilizer. Males, however, were given a mixture of etorphine (4.5 mg), medetomidine (4.5 mg), and ketamine (200 mg), administered intramuscularly via non-metal dart. This information has now been added to the Study Design and Data Collection (Page 8, Line 142 and Page 8, Line 150). 

Page 8, Line 142: No chemical immobilizations were used.

Page 8, Line 150: Males were immobilized with a mixture of etorphine, medetomidine, and ketamine.

Line 138: Please provide photo of animal within the TAMER

Response: A picture of the TAMER is provided as supplemental information for the reviewer, but not meant to be included in the manuscript. We have, admittingly, poor pictures of the TAMER in action. Better pictures of the TAMER can be found on the Fauna Research Incorporated website (https://www.faunaresearch.com/).

Line 139: Could the procedure of AI have impacted FGM readings?

Response: AI injections would not impact the FGM response because they are too short-term of a stressor to register a response and because closprostenol sodium does not cross-react with the antibody used in the RIA analysis. In addition, similar to our answer below on potential confounding effects from semen collection, we are careful not to overstate our conclusions. In the Concluding paragraph (Page 33, Line 691), we state thatwe are unable to fully separate the effects of capture from those of the GPS collar. For our FGM analyses, we included 2 female scimitar-horned oryx as control animals. Neither animal received injections for AI, nor were they fitted with a GPS collar. Each animal, however, was subject to similar capture-related stress activities. Our results show that the stress of these animals remained constant over the study period (i.e., no change). Further evaluation to isolate these confounding factors (animal capture vs GPS collar) was not conducted and further work on this would be ideal. Our analyses of accelerometry, however, do clearly indicate that headshaking, a behavior that is likely only attributable to the GPS collar over a long period, increased immediately after animal capture/GPS collar fitting. Importantly, this elevated response in headshaking can still be seen in animals the day after the collars were fitted, a response most likely to the GPS collar and not the capture. 

Line 145: Semen collection is a standard procedure which may generate physiological stress in wildlife. It is unclear if the experimental design for testing the effects of GPS collaring is correct if other procedures were done in combination and it makes it difficult to tease apart the cause-effects?

Response: Three male scimitar-horned oryx were included in our study, two of which were anesthetized using a combination of etorphine, medetomidine, and ketamine and subjected to electroejaculation for semen collection. The third male, 113204, was included as a control for FGM analyses. This is, admittingly, a small sample size. Together with the two female scimitar-horned oryx that were also used as controls, our results show that FGM stress levels remained unchanged across treatment periods. Disentangling the effect from capture and those that result directly from semen collection, however, is inherently difficult because one needs to capture the animal first, before you can collect semen. We are unable to find any scientific literature that separates these two effects, but since animals were anesthetized during electroejaculation, it would seem that semen collection would be a minor confounding effect. We do not, however, have any concrete data to support this claim. We have been careful throughout the manuscript to not overstate our conclusions, given the potential confounding effect with animal capture, and state explicitly in the concluding paragraph (Page 33, Line 691) that “we are unable to disentangle the effects of animal handling from the effects related solely to GPS collars themselves”. At the same time, we use multiple forms of evidence, one of which (accelerometer analysis of headshaking) is very specific to the effects of the GPS collar only. Further, while we can’t disentangle effects of the collar from effects of the capture, these items will always come as a ‘package’. Thus, the short-term effects of collaring shown in our study are then inclusive of any effects of handling (as they should be). 

Line 175: please verify how the behaviour aspect presented in the current manuscript is different from the recent publication (BioRxiv)?

Response: There is no difference between the manuscript submitted here for review and the preprint that appears on BioRxiv (i.e., these documents are identical). PlosOne offered the option to publish a preprint at the time the manuscript was submitted for review.

Line 251: When were the assays performed since sample collection in 2015?

Response: Assays were performed shortly after samples were collected. The last fecal samples were collected on 18 December 2015. All assays were completed within 1-month of the last samples collected (18 January 2016). Information added to Data Collection and Processing.

Page 14, Line 271: Samples were homogenized during collection by hand to more evenly distribute hormones and decrease sample variability and assayed within 1-month of the last date of sample collection.

Line 255: It is unclear why corticosterone instead of cortisol (a major glucocorticoid of mammals) was measured?

Response: Correct. Cortisol is a major glucocorticoid hormone in mammals and is often found in higher concentrations than corticosterone. However, we used a corticosterone assay to measure glucocorticoid metabolites in the feces because little native cortisol is excreted. While some cortisol assays may detect some cortisol, the corticosterone RIA has by far the best cross reactivity. Thus, we measured the excreted metabolites using a broad spectrum corticosterone antibody. No change made to text. 

Line 257: Please provide the sensitivity of the RIA.

Response: The sensitivity (1 ng/mL) of the RIA has been added to the Methods section.

Page 14, Line 277: The sensitivity of the assay is 1 ng/mL

Line 267: Is this hypothesis correct if animals were sampled for semen and females were AI’ed during collaring?

Response: Please see responses regarding semen collection and AI during collaring provided above. Semen collection and/or AI would not be expected to impact fecal glucocorticoid analyses. 

Line 282: As the gut-passage time been biologically validated in the study species? Please provide a reference for the gut-passage time

 Response: Apologies. Reference were provided in the text, but not included in the text of the Figure. We have now added Warner 1981, which isn’t specific to scimitar-horned oryx, but does provide examples from a variety of ruminants that would be expected to have similar retention times as oryx (~ 1 day). Figure and text now updated with: 

Schwarzenberger F. The many uses of non-invasive faecal steroid monitoring in zoo and wildlife species. Int Zoo Yearb. 2007;41: 52–74. doi:10.1111/j.1748-1090.2007.00017.x

Hodges K, Brown J, Heistermann M. Endocrine Monitoring of Reproduction and Stress. In: Kleiman DG, Thompson K V., Kirk Baer C, editors. Wild Mammals in Captivity: Principles and Techniques for Zoo Management. Chicago: University of Chicago Press; 2010. pp. 447–468. 

Warner, A.C.I. 1981. Rate of passage of digesta through the gut of mammals and birds. Nutrition Abstracts and Reviews. 51(12): 789-820.

---

## [Editor Report · Decision Letter 1]

22 Jan 2020

Short-term effects of GPS collars on the activity, behavior, and adrenal response of scimitar-horned oryx (Oryx dammah)

PONE-D-19-23204R1

Dear Dr. Stabach,

We are pleased to inform you that your manuscript has been judged scientifically suitable for publication and will be formally accepted for publication once it complies with all outstanding technical requirements.

With kind regards,

Edna Hillmann, Ph.D.

Academic Editor

PLOS ONE

Additional Editor Comments (optional):

Reviewers’ comments:

---

## [Editor Report · Acceptance letter]

31 Jan 2020

PONE-D-19-23204R1 

Short-term effects of GPS collars on the activity, behavior, and adrenal response of scimitar-horned oryx (*Oryx dammah*) 

Dear Dr. Stabach:

I am pleased to inform you that your manuscript has been deemed suitable for publication in PLOS ONE. Congratulations! Your manuscript is now with our production department. 

With kind regards,

on behalf of

Dr. Edna Hillmann 

Academic Editor

PLOS ONE